# Risk factors of furazolidone-associated fever

Jiali Zhang[1], Chunling Rong[1], Chenyang Yan[2], Jie Chen[1], Wenjun Yang[1], Lingyan Yu[1], Haibin Dai[1]*

**1** Department of Pharmacy, Second Affiliated Hospital, Zhejiang University School of Medicine, Hangzhou, China, **2** Department of Pharmacy, Marine Police Hospital, Jiaxing, China

* haibindai@zju.edu.cn

**Data Availability Statement:** The data that support the findings of this study contain potentially sensitive patient information. The Committee of the 2nd Affiliated Hospital, School of Medicine, Zhejiang University has imposed the restrictions

## Abstract

### Background

Furazolidone is a synthetic nitrofuran with a broad spectrum of antimicrobial action and has been widely used in the treatment of *Helicobacter pylori (H. pylori)* infection. However, its safety profile has not been clarified. Moreover, the drug fever associated with its use is frequently misdiagnosed. The aim of this study was to explore the risk factors of furazolidone-associated fever to increase awareness and stimulate further research on this topic.

### Methods

This was a retrospective case-control study of patients referred to a specialist clinic for furazolidone-containing quadruple regimens for *H. pylori* infection at a tertiary care hospital located in Eastern China between July 2018 and September 2018. We evaluated adult patients who received furazolidone treatment for *Helicobacter pylori* infection. The exclusion criteria were as follows: (1) patients were pregnant or breastfeeding; (2) patients received furazolidone treatment not for *Helicobacter pylori* infection; (3) patients had taken antibiotics or any acid suppressant or non-steroidal anti-inflammatory drug in the last 4 weeks; (4) patients had chronic hepatic, renal, or pulmonary disease. Pertinent information was retrieved from medical records and telephone follow-up. All statistical analysis was performed in SPSS version 22.0.

### Results

A total of 1499 patients received furazolidone and met the overall inclusion criterion. Of these 1499 patients, 27 (1.80%) developed drug fever. The mean time between initiation of furazolidone and the onset of fever is 11.00 ± 1.84 days, and the median peak fever was 38.87 ± 0.57˚C. We found no differences in age and past drug allergy between the non-fever and fever groups. Through multiple logistic regression analysis, we found two variables as independent risk factors for furazolidone-associated fever, including gender (OR, 3.16; 95% CI, 1.26–7.91; *P* = 0.014) and clarithromycin (OR, 4.83; 95% CI, 2.17–10.79; *P*<0.001).

### Conclusions

This retrospective cohort study identified two risk factors for furazolidone-associated fever, which were female and clarithromycin. We also analyzed the characteristics of drug fever

apply to the availability of these data, which were used under license for this study. Data are available from corresponding author Haibin Dai with the permission of the Ethics Committee of the 2nd Affiliated Hospital, School of Medicine, Zhejiang University. The contact information of the Ethics Committee is keyanlunli_zheer@163.com.

**Funding:** Yes, this project was supported by a grant from the Natural Science Foundation of Zhejiang Province (Grant No. LYY19H310013) which was received by JL Zhang, the Foundation from the Health Bureau of Zhejiang Province (Grant No. 2019KY406) which was received by JL Zhang, and the foundation from Zhejiang Pharmaceutical Association (Grant No. 2019ZYY18) which was received by LY Yu.

**Competing interests:** The authors have declared that no competing interests exist.

during anti-*Helicobacter pylori* therapy. However, the underlying mechanisms are uncertain and require further research.

## Introduction

*Helicobacter pylori (H. pylori)* infection is common worldwide, with a prevalence of 18.9% and 79.1% in adults [1]. The most commonly recommended therapies worldwide consist of a proton pump inhibitor (PPI), clarithromycin, amoxicillin, and/or metronidazole [2]. Many studies have validated that antibiotic resistance is the clear leading cause of treatment failure [3]. Recently, a review published by Hu et al. revealed that clarithromycin, metronidazole, and levofloxacin resistance rates in Chinese patients were 28.9%, 63.8%, and 28.0%, respectively [4]. However, the resistance rates of other antibiotics, such as tetracycline and furazolidone, remain low [5]. Studies have shown that the *H. pylori* eradication rates following furazolidone-containing therapies are higher than 80% [6–8]. Because of its high eradication rates and low cost, the Fifth Chinese National Consensus Report recommended that furazolidone should be used for *H. pylori* eradication treatment [9]. Furazolidone is chemically related to nitrofurantoin, which has primarily been used in humans for the treatment of diarrheal diseases [10]. The most common adverse effects of furazolidone include gastrointestinal reactions, such as nausea, vomiting, abdominal pain and headache, as well as allergic reactions [11, 12]. Fever was also, by far, one of the most commonly observed side effects, which are major determinants of compliance. In a retrospective review by Altamirano and Bondani, of 10443 patients who were treated with furazolidone, approximately 0.34% experienced fever [13]. Drug fever accounts for 0.01%-5% of all adverse reactions to drugs and is a common condition that is frequently misdiagnosed [14, 15]. A wide variety of agents have been shown to induce drug fever [15]. Among such agents, antimicrobials are the most common causes of drug fever [16]. In the past few years, furazolidone has been chosen as a treatment because of the lack of resistance of *H. pylori* against this drug, as well as its high efficacy [17, 18]. However, its safety remains inconclusive; the incidence of drug fever in clinical findings was found to be much higher than that reported. Overall, it can be concluded that the inclusion of furazolidone in a treatment regimen for infection does not appear to be absolutely safe [19].

In the present study, 1499 patients who received furazolidone treatment for *H. pylori* infection were analyzed to summarize the clinical features of drug fever caused by furazolidone. These findings may improve the diagnosis and treatment of drug fever associated with furazolidone. Our goal is to increase awareness and stimulate further research on this topic.

## Materials and methods

### Study design

This was a retrospective case-control study using the medical records of patients treated with 14-day Furazolidone-containing quadruple regimens for *Helicobacter pylori* (*H. pylori*) infection at the Second Affiliated Hospital of Zhejiang University, School of Medicine (SAHZU), which is a tertiary care hospital located in Eastern China, with a total capacity of 3200 licensed beds. The enrolled participants were infected between July 2018 and September 2018 and were all of Han Chinese ethnicity. Participants were all anonymous, and investigators had no access to patient information during or after data collection. The inclusion criteria selected (1) patients aged ≥18 years and (2) patients who received furazolidone treatment for *Helicobacter pylori* infection. The following patients were excluded: (1) patients who were pregnant or

breastfeeding; (2) patients who received furazolidone treatment not for *Helicobacter pylori* infection; (3) patients taking antibiotics or any acid suppressant or non-steroidal anti-inflammatory drug in the last 4 weeks; and (4) patients with chronic hepatic, renal, or pulmonary disease.

## Data collection and definitions

The possibility of a specific drug causing fever was evaluated by Naranjo algorithm [20]. Two independent reviewers gave each suspected patients with drug fever a Naranjo scale score. The scores were averaged and rounded up to the higher integer. The final score interpretations were stratified into four categories with a score of $\geq 9$ considered "definite", 5 to 8 "probable", 1 to 4 "possible", and those $\leq 0$ "doubtful" likelihood of the drug causing the ADR. Fever was defined as body temperature $\geq 38°C$. It was concluded that fever had been caused by the medication if it cleared rapidly after the suspected drug was discontinued [21]. We conducted a follow-up assessment of these patients for one month after they left hospital.

For each patient, the collected data included demographics such as age, gender, weight, and past drug allergy. We also collected data on the duration of the regimen and concomitant drugs (e.g., amoxicillin, clarithromycin, levofloxacin, colloidal bismuth pectin, bismuth potassium citrate, pantoprazole, rabeprazole, esomeprazole, and lansoprazole) with furazolidone therapy in patients with drug fever. In addition, white blood cells, neutrophils and eosnophils were obtained from the lab results.

## Statistical analysis

Continuous variables are presented as the mean ± SD or median (interquartile range [IQR]) depending on whether they are normally distributed. Categorical variables are expressed as frequencies (%). Using Student's t-tests for independent variables and the Mann–Whitney U test for non-normally distributed data, we compared continuous variables. In contrast, Pearson's $\chi^2$-test was used to compare categorical variables. Separate logistic regression analyses and multivariable logistic regression analyses were performed to determine risk factors associated with drug fever during anti-*Helicobacter pylori* therapy. Covariates were based on significant variables in the univariable model (i.e., $P < 0.05$). All statistical analyses were performed in SPSS for Windows version 22.0. $P < 0.05$ was considered statistically significant.

## Ethics approval

This study was approved by the medical ethics committee of the Second Affiliated Hospital, Zhejiang University School of Medicine, China (No. 2020–283). The need for informed consent was waived due to the retrospective design of this study. All data collected was kept confidential.

## Results

A total of 1503 patients were enrolled in the study. However, 4 patients were later excluded because they met the exclusion criteria, which included receiving furazolidone treatment for a reason other than *Helicobacter pylori* infection (n = 3) and taking antibiotics or any acid suppressant or non-steroidal anti-inflammatory drug in the last 4 weeks (n = 1). Eventually, 1499 patients were included, and 27 of these patients showed furazolidone drug fever. The Naranjo score classified 0 (0.0%) of cases as definite, 25 (92.6%) probable, 2 (7.4%) possible, and 0 (0.0%) doubtful (Table 1). The incidence of furazolidone drug fever was as high as 1.80%. The average age of the adverse reaction group was 38.85±11.34 years old, and the non-fever group

**Table 1. Naranjo score of patients with observed fever.**

| Naranjo score | N (%) |
|---|---|
| Definite ($\geq 9$) | 0 (0.0) |
| Probable (5–8) | 25 (92.6) |
| Possible (1–4) | 2 (7.4) |
| Doubtful ($\leq 0$) | 0 (0.0) |

was 39.94±12.66 years old. Notably, we found no differences in age and past drug allergy between the non-fever and fever groups. Among the 27 fever patients, 21 cases were female, accounting for 77.8%, which was significantly different between the two groups ($P = 0.005$). The average body weight of the fever group was 57.52±7.60kg, and that of the non-fever group was 62.6±13.90kg, indicating a significant difference between the two groups ($P = 0.037$) (Table 2). Concomitant medications, except for amoxicillin and clarithromycin ($P = 0.004$ and $P<0.001$, respectively), were similar between the groups (Table 3).

As shown in Table 4, the mean time between the initiation of furazolidone and the onset of fever is 11.00±1.84 days, and the median peak fever was 38.87±0.57˚C. The leukocyte count and eosinophil level were both normal. The neutrophil mean count was $(8.34±3.57) \times 10^9$/L, which was slightly elevated in this research.

Univariate analysis revealed four variables related to drug fever, including gender, weight, amoxicillin, and clarithromycin. Furthermore, we revealed two independent variables to be associated with the onset of drug fever after adjusting the underlying confounders through multivariate analysis. These variables included gender (OR, 3.162; 95% CI, 1.264–7.914; $P = 0.014$) and clarithromycin (OR, 4.834; 95% CI, 2.165–10.794; $P<0.001$) (Table 5).

## Discussion

Although known as a rare adverse drug reaction (ADR), drug fever remains an important issue in medicine, with the risk of leading to inappropriate and potentially harmful diagnostic and therapeutic interventions. Only sparse data regarding drug fever have been published. In this case series, we present 27 (1.80%) patients with drug fever out of 1499 patients administered furazolidone during a three-month period in our hospital. Furazolidone is a synthetic nitrofuran with a broad spectrum of antimicrobial action and has been widely used in the treatment of gastrointestinal infections [11]. The side effects of furazolidone-containing therapies vary across different studies, and they occur more frequently with these therapies than with standard therapies. Side effects include nausea, vomiting, diarrhea, drug fever, skin rash, and so on [22, 23]. In this study, the incidence of drug fever caused by furazolidone was 1.80%, which was higher than 0.34% reported in previous assessments [13]. Drug fever may occur at any point during a course of drug therapy, and there is significant variation among different

**Table 2. Demographic and clinical characteristics of patients with *Helicobacter pylori* infection.**

| Variable | Non-fever(n = 1472) | Fever(n = 27) | P |
|---|---|---|---|
| Age, years | 39.94±12.66 | 38.85±11.34 | 0.237 |
| Gender, male | 718(48.8%) | 6(22.2%) | 0.005 |
| Gender, female | 754(51.2%) | 21(77.8%) | 0.005 |
| Weight, kg | 62.6±13.90 | 57.52±7.60 | 0.037 |
| Past drug allergy | 150(10.2%) | 2(7.4%) | 0.473 |

Data are presented as n (%) or mean ± SD

**Table 3. Drug dosing and concomitant drugs in patients with or without Furazolidone-associated fever.**

| Furazolidone combined drugs | Non-fever(n = 1472) | Fever(n = 27) | P |
|---|---|---|---|
| Amoxicillin (1.0g bid) | 1256(85.3%) | 17(63.0%) | **0.004** |
| Clarithromycin (0.5g bid) | 152(10.3%) | 10(37.0%) | **< 0.001** |
| Levofloxacin (0.5g qd) | 39(2.6%) | 0 | 0.488 |
| Colloidal bismuth pectin (0.2g bid) | 889(60.4%) | 15(55.6%) | 0.374 |
| Bismuth potassium citrate (0.6g bid) | 574(39.0%) | 12(44.4%) | 0.349 |
| Pantoprazole (40mg bid) | 336(22.8%) | 4(14.8%) | 0.232 |
| Rabeprazole (10mg bid) | 807(54.8%) | 19(70.4%) | 0.077 |
| Esomeprazole (20mg bid) | 201(13.7%) | 3(11.1%) | 0.488 |
| Lansoprazole (30mg bid) | 121(8.2%) | 2(7.4%) | 0.616 |

Data are presented as n (%).

drug classes. Drug fever most commonly appears after 7–10 days of drug administration [15]. In this study, the average cumulative days of furazolidone use in patients with fever was 11.00 ±1.84 days, which was slightly longer than that found for other antibacterial drugs. According to Johnson D.H. and Cunha, laboratory findings can be helpful in supporting a diagnosis of drug fever, although they are highly variable and cannot be relied on for a definitive diagnosis [24]. Leukocytosis with or without a left shift may be present. In all patients with suspected drug fever in our study, the leukocyte count was within the normal range. Previous studies have shown that eosinophil is increased in patients with suspected drug fever. However, this was not the case in our study. No other laboratory tests were carried out in this study. Once furazolidone was discontinued, the body temperature returned to normal lever after 48 h. Furazolidone interferes with the activity of bacterial oxidoreductase, blocking bacterial metabolism. However, the exact mechanisms of furazolidone that cause fever are remain undetermined. However, the drug is chemically related to nitrofurantoin, which is well known to cause pleuropneumonic reactions [25]. These acute hypersensitivity reactions typically begin with fever, cough, and dyspnea [26]. The proposed mechanisms involve a cytotoxic response, an immune-complex mediated response and a cell mediated reaction [27]. The mechanism of pyrexia in cellular immunity appears to be due to production of nonpyrogenic-soluble factors that act on macrophages to produce endogenous pyrogen, thereby resulting in fever [15]. On the other hand, as early as the 1960s, furazolidone was found to be an inhibitor of monoamine oxidase activity. It can interact with a number of drugs and commonly consumed foods, such as soy sauce, aged cheese, and beer [28]. For patients taking furazolidone, the consumption of foods containing tyramine may stimulate the sympathetic nerves. This can be prevented by avoiding the consumption of tyramine-containing foods while receiving furazolidone treatment. In addition, during furazolidone therapy a small amount of tea or coffee can cause

**Table 4. Duration of regimens and laboratory tests of patients with observed fever.**

| Laboratory Tests | Mean | Normal Ranges |
|---|---|---|
| Duration of regimen (days) | 11.00±1.84 | |
| Body temperature (°C) | 38.87±0.57 | 36.0~37.0 |
| White blood cells (×10$^9$/L) | 9.84±3.63 | 4.0~10.0 |
| Neutrophils (×10$^9$/L) | 8.34±3.57 | 2.0~7.0 |
| Eosnophils (×10$^9$/L) | 0.18±0.15 | 0.0~1.0 |

Data are presented as mean ± SD.

**Table 5. Univariable and multivariable logistic regression analysis for independent risk factors for furazolidone-associated fever in patients.**

| Risk factor | Unadjusted OR (95% CI) | P | Adjusted OR (95% CI) | P |
|---|---|---|---|---|
| Gender | 3.333(1.338–8.305) | 0.010 | 3.162(1.264–7.914) | 0.014 |
| Weight | 0.967(0.937–0.999) | 0.045 | | |
| Amoxicillin | 0.292(0.132–0.647) | 0.002 | | |
| Clarithromycin | 5.108(2.298–11.357) | < 0.001 | 4.834(2.165–10.794) | < 0.001 |

OR, odd ratio; 95% CI, 95% confidence interval.

insomnia, and a large amount of tea can trigger high blood pressure. Therefore, tea or coffee should not be consumed less than two hours before taking furazolidone.

To the best of our knowledge, this is the first report on the independent risk factors for furazolidone-associated drug fever. In this single-center study evaluating furazolidone-based quadruple therapy for *H. pylori* infection, the dosage of furazolidone was 0.1 g, b.i.d., according to the drug instruction. In addition, Roghani et al. reported that fever, fatigue, and dizziness were more common in high-dose furazolidone-containing therapies (200 mg, b.i.d.) than low-dose therapies (50 mg, b.i.d.) [29]. These results indicate that the side-effects of furazolidone are associated with dose. While there was no difference in furazolidone dose (100 mg, b.i.d.) between the two groups in our study, we cannot determine whether furazolidone dose influenced drug fever. Other investigated risk factors including age, weight, past drug allergy, and concomitant administration for *H. pylori* infection, except for clarithromycin, were similar between the two groups. Retrospectively, we noted that the drug fever of furazolidone was significantly associated with gender (female versus male, *P*=0.014). Lipsky and Hirschmann reported that the risk of drug fever is significantly increased in female patients [30], which is consistent with the results of this study. A highlight of this study was that clarithromycin was revealed to be an independent risk factor for drug fever (*P*<0.001). In this study, the combination of furazolidone and amoxicillin was not prone to drug fever, but the combined use of clarithromycin was more prone to drug fever. The inhibition of hepatic cytochrome P-450 enzymes by clarithromycin has been well documented [31]. The mechanism of inhibition probably involves the induction of demethylation and oxidation by cytochrome P-450 of the tertiary amine on the amino sugar of the macrolide to its nitroso intermediate, which then forms an inactive complex with the iron of cytochrome P-450. Reported drug interactions include enhanced glucocorticoid effects, theophylline toxicity, carbamazepine toxicity, and the clinical failure of oral contraceptives [32]. Whether it will affect the metabolism of furazolidone has not been reported yet, and further studies are needed.

This study suffers from the following limitations. Firstly, it was a retrospective single-center study with a small sample size. Our 27 patients represent a convenience sample; the true incidence of furazolidone-associated drug fever cannot be determined with certainty. Secondly, drug fever may have been induced by multifactorial causes including diet, comorbidities, age, weight, drug–drug interactions, and genetics. Individual genetic variation in key genes involved in the metabolism, drug transport, or drug target can contribute to the risk of adverse events. Although we examined various factors related to furazolidone-associated drug fever, some other risk factors that may cause this adverse effect to remain unexamined. Thirdly, as all patients were outpatients in this study, there were very few laboratory values on them. We could not collect enough data due to the limitations of our design. This should be considered in further studies.

In summary, we described 27 patients infected with *H. pylori* with the onset of clinically relevant hyperthermia during furazolidone administration, highly suggestive of drug fever. We

also identified being female and the combined use of clarithromycin as risk factors for furazolidone-induced drug fever. Drug fever is a common and potentially serious adverse reaction to furazolidone of which clinicians should have increased awareness. As the underlying mechanisms are uncertain to date, further research on this topic is warranted.

## Acknowledgments

We wish to thank Linxiao Dai, Xiuping Zhu, Huan Luo, Yuwen Huang, Wei He, Junfeng Li for their significant help with this work.

## Author Contributions

**Conceptualization:** Jiali Zhang, Haibin Dai.

**Data curation:** Jiali Zhang, Chunling Rong, Chenyang Yan.

**Formal analysis:** Jie Chen.

**Funding acquisition:** Jiali Zhang, Lingyan Yu.

**Investigation:** Jiali Zhang.

**Methodology:** Jie Chen, Wenjun Yang, Lingyan Yu.

**Supervision:** Haibin Dai.

**Validation:** Haibin Dai.

**Writing – original draft:** Jiali Zhang.

**Writing – review & editing:** Chunling Rong, Chenyang Yan, Jie Chen, Wenjun Yang, Lingyan Yu.

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
