## [Decision Letter · Decision Letter 0]

25 Oct 2021

PONE-D-21-28795Risk Factors of Furazolidone-Associated FeverPLOS ONE

Dear Dr. Dai

Thank you for submitting your manuscript to PLOS ONE. After careful consideration, we feel that it has merit but does not fully meet PLOS ONE’s publication criteria as it currently stands. Therefore, we invite you to submit a revised version of the manuscript that addresses the points raised during the review process.

We look forward to receiving your revised manuscript.

Kind regards,

Muhammad Shahzad Aslam, Ph.D.,M.Phil., Pharm-D

Academic Editor

PLOS ONE

Journal Requirements:

“This project was supported by a grant from the Natural Science Foundation of Zhejiang Province (Grant No. LYY19H310013), the Foundation from the Health Bureau of Zhejiang Province (Grant No. 2019KY406), and the foundation from Zhejiang Pharmaceutical Association (Grant No. 2019ZYY18).”

We note that you have provided additional information within the Funding Section. Please note that funding information should not appear in other areas of your manuscript. We will only publish funding information present in the Funding Statement section of the online submission form.

“Yes, this project was supported by a grant from the Natural Science Foundation of Zhejiang Province (Grant No. LYY19H310013) which was received by JL Zhang, the Foundation from the Health Bureau of Zhejiang Province (Grant No. 2019KY406) which was received by JL Zhang, and the foundation from Zhejiang Pharmaceutical Association (Grant No. 2019ZYY18) which was received by LY Yu.”

4. Thank you for stating the following in the Competing Interests/Financial Disclosure * (delete as necessary) section:

“This project was supported by a grant from the Natural Science Foundation of Zhejiang Province (Grant No. LYY19H310013), the Foundation from the Health Bureau of Zhejiang Province (Grant No. 2019KY406), and the foundation from Zhejiang Pharmaceutical Association (Grant No. 2019ZYY18).”

We note that you received funding from a commercial source: Zhejiang Pharmaceutical Association

Reviewers' comments:

Reviewer's Responses to Questions

**Comments to the Author**

1. Is the manuscript technically sound, and do the data support the conclusions?

Reviewer #1: Yes

Reviewer #2: Partly

2. Has the statistical analysis been performed appropriately and rigorously? 

Reviewer #1: Yes

Reviewer #2: I Don't Know

3. Have the authors made all data underlying the findings in their manuscript fully available?

Reviewer #1: Yes

Reviewer #2: No

4. Is the manuscript presented in an intelligible fashion and written in standard English?

Reviewer #1: Yes

Reviewer #2: No

5. Review Comments to the Author

Reviewer #1: The underlying mechanisms of furazolidone causing fever should be discussed.

Line#64 However, the resistance rates of other 64 antibiotics, such as tetracycline and furazolidone, are still low. Reference is missing.

Line#77 A wide 77 variety of agents can cause drug fever Reference is missing.

Line# 82 Overall, the 82 inclusion of furazolidone in a treatment regimen for infection does not absolutely 83 appear to be safe

Reference is missing.

How the authors conducted a follow up on the study patients for one month after they left hospital?

The average age of the adverse reaction group was 38.85±11.34 years old, and the other group was 39.94±12.66 years old. The sentence is NOT clear what is the other group?

Reviewer #2: i have attached the document with my comments for detail please check it

Explain the exclusion criteria; for ) patients were pregnant or breastfeeding- as being the high risk population why they were excluded

in abstract - result section mentioned -A total of 1499 patients received polymyxin and met the overall inclusion criterion- may be had to write Furazolidone and by mistake mentioned polymyxin - needs correction

Drug response can be impacted by several factors including diet, comorbidities, age, weight, drug–drug interactions, and genetics. Individual genetic variation in key genes involved in the metabolism, transport, or drug target can contribute to risk of adverse events or treatment failure. I feel very little interest while reading this paper as very important aspects are missing.

The manuscript should be checked by a native speaker in terms of language and grammar. I highlighted only some passages in the attached document.

I am not sure for PLOS ONE criteria for manuscript selection but the finding of the paper are not significant due to lack of data about very important variables - that can address this topic clearly

6. PLOS authors have the option to publish the peer review history of their article (what does this mean?). If published, this will include your full peer review and any attached files.

Reviewer #1: **Yes: **Dr. Sadia Shakeel

Reviewer #2: **Yes: **Dr. Gul Ambreen

---

## [Author Response · Author response to Decision Letter 0]

7 Dec 2021

Thank you for your letter and for the reviewers' comments concerning our manuscript entitled “Risk Factors of Furazolidone-Associated Fever”. These comments are all valuable and very helpful for revising and improving our paper, as well as the important guiding significance to our research. We have studied comments carefully and have made corrections which we hope to meet with approval. The responds to the editor and reviewers' comments are as follows:

Journal Requirements:

Response: Thank you for your suggestion. We have read the PLOS ONE style templates and modified one by one according to it.

Response: Thank you for your suggestion. The data that support the findings of this study contain potentially sensitive patient information. Restrictions apply to the availability of these data, which were used under license for this study. Data are available from corresponding author Haibin Dai with the permission of the Ethics Committee of the 2nd Affiliated Hospital, School of Medicine, Zhejiang University. 

 We have revised cover letter and explain the reasons of ethical restrictions on sharing a de-identified data set. We also provide contact information which data requests may be sent in the revised cover letter. 

“This project was supported by a grant from the Natural Science Foundation of Zhejiang Province (Grant No. LYY19H310013), the Foundation from the Health Bureau of Zhejiang Province (Grant No. 2019KY406), and the foundation from Zhejiang Pharmaceutical Association (Grant No. 2019ZYY18).”

We note that you have provided additional information within the Funding Section. Please note that funding information should not appear in other areas of your manuscript. We will only publish funding information present in the Funding Statement section of the online submission form.

“Yes, this project was supported by a grant from the Natural Science Foundation of Zhejiang Province (Grant No. LYY19H310013) which was received by JL Zhang, the Foundation from the Health Bureau of Zhejiang Province (Grant No. 2019KY406) which was received by JL Zhang, and the foundation from Zhejiang Pharmaceutical Association (Grant No. 2019ZYY18) which was received by LY Yu.”

Response: Thank you for your suggestion. We have removed the funding-related text from the manuscript. My funding statement is the same as before: “This project was supported by a grant from the Natural Science Foundation of Zhejiang Province (Grant No. LYY19H310013) which was received by JL Zhang, the Foundation from the Health Bureau of Zhejiang Province (Grant No. 2019KY406) which was received by JL Zhang, and the foundation from Zhejiang Pharmaceutical Association (Grant No. 2019ZYY18) which was received by LY Yu.” 

4. Thank you for stating the following in the Competing Interests/Financial Disclosure * (delete as necessary) section:

“This project was supported by a grant from the Natural Science Foundation of Zhejiang Province (Grant No. LYY19H310013), the Foundation from the Health Bureau of Zhejiang Province (Grant No. 2019KY406), and the foundation from Zhejiang Pharmaceutical Association (Grant No. 2019ZYY18).”

We note that you received funding from a commercial source: Zhejiang Pharmaceutical Association

Response: Thank you for your suggestion. Zhejiang Pharmaceutical Association is a none profit organization in China. The founder from Zhejiang Pharmaceutical Association is also not a commercial funder. We adhere PLOS ONE policies on sharing data and materials all the time. 

Reviewers' comments:

1. Is the manuscript technically sound, and do the data support the conclusions?

Reviewer #1: Yes

Reviewer #2: Partly

Response: Yes, our manuscript described a technically sound piece of scientific research. The research has been conducted rigorously, with appropriate controls and sample sizes. This retrospective cohort study identified two risk factors for furazolidone-associated fever, which were female and clarithromycin. We also analyzed the characteristics of drug fever during anti-Helicobacter pylori therapy. These conclusions were drawn appropriately based on the data presented. 

2. Has the statistical analysis been performed appropriately and rigorously?

Reviewer #1: Yes

Reviewer #2: I Don't Know

Response: Yes, the statistical analysis has been performed appropriately and rigorously with SPSS for Windows version 22.0 in our manuscript. 

3. Have the authors made all data underlying the findings in their manuscript fully available?

Reviewer #1: Yes

Reviewer #2: No

Response: Yes, the data that support the findings of this study contain potentially sensitive patient information. Data are available from corresponding author Haibin Dai with the permission of the Ethics Committee of the 2nd Affiliated Hospital, School of Medicine, Zhejiang University. 

4. Is the manuscript presented in an intelligible fashion and written in standard English?

Reviewer #1: Yes

Reviewer #2: No

Response: Yes, the manuscript has been thoroughly revised and edited by a native speaker. The certificate provided by multidisciplinary digital publishing institute was in the attached document. We really hope that the language level have been substantially improved. 

Reviewer #1:

The underlying mechanisms of furazolidone causing fever should be discussed.

Response: Thank you for your suggestion. We have added the following discussion to further explain the underlying mechanisms of furazolidone causing fever in revised manuscript as follow: “Line#201-211 However, the exact mechanisms of furazolidone causing fever are remain undetermined. It is chemically related to nitrofurantoin, which is well known to cause pleuropneumonic reactions [25]. These acute hypersensitivity reactions typically begin with fever, cough, and dyspnea [26]. The proposed mechanisms involve a cytotoxic response, an immune-complex mediated response and a cell mediated reaction [27]. The mechanism of pyrexia in cellular immunity appears to be due to production of nonpyrogenic soluble factors that act on macrophages to produce endogenous pyrogen to result in fever [15]. On the other hand, as early as the 1960s, furazolidone was shown to be an inhibitor of monoamine oxidase activity. It can interact with a number of drugs and commonly used foods such as soy sauce, aged cheese, and beer [28].” 

Line#64 However, the resistance rates of other antibiotics, such as tetracycline and furazolidone, are still low. Reference is missing.

Response: Thank you for your suggestion. We have cited the reference “However, the resistance rates of other antibiotics, such as tetracycline and furazolidone, are still low [5].”

Line#77 A wide variety of agents can cause drug fever Reference is missing.

Response: Thank you for your suggestion. We have cited the reference “A wide variety of agents have been shown to induce drug fever [15].” 

Line# 82 Overall, the inclusion of furazolidone in a treatment regimen for infection does not absolutely appear to be safe. Reference is missing. 

Response: Thank you for your suggestion. We have cited the reference “Overall, the inclusion of furazolidone in a treatment regimen for infection does not absolutely appear to be safe [19].” 

How the authors conducted a follow up on the study patients for one month after they left hospital? 

Response: Thank you for your suggestion. Patients were contacted by telephone at 0.5 and 1 months after they left hospital by pharmacists who were blinded to the grouping. 

The average age of the adverse reaction group was 38.85±11.34 years old, and the other group was 39.94±12.66 years old. The sentence is NOT clear what is the other group?

Response: Thank you for your suggestion. We are very sorry for the inconvenience it caused in your reading. We have changed the sentence “The average age of the adverse reaction group was 38.85±11.34 years old, and the other group was 39.94±12.66 years old” to “The average age of the adverse reaction group was 38.85±11.34 years old, and the non-fever group was 39.94±12.66 years old” in the revised manuscript. 

Reviewer #2:

Explain the exclusion criteria; for) patients were pregnant or breastfeeding- as being the high risk population why they were excluded

Response: Thank you for your suggestion. Furazolidone is contraindicated in the dispensatory for pregnant and breastfeeding women in China. Furazolidone used by pregnant or breastfeeding patients is an off-label use. However, none of the 4 patients excluded in this study were pregnant or breastfeeding women. 

in abstract - result section mentioned -A total of 1499 patients received polymyxin and met the overall inclusion criterion- may be had to write Furazolidone and by mistake mentioned polymyxin - needs correction

Response: We are extremely grateful to you for pointing out this problem. We have changed this sentence to “A total of 1499 patients received furazolidone and met the overall inclusion criterion”. 

Drug response can be impacted by several factors including diet, comorbidities, age, weight, drug–drug interactions, and genetics. Individual genetic variation in key genes involved in the metabolism, transport, or drug target can contribute to risk of adverse events or treatment failure. I feel very little interest while reading this paper as very important aspects are missing.

Response: Thank you for your suggestion. We agree that drug response can be impacted by several factors. In fact, our research is the first to show the independent risk factors for furazolidone-associated drug fever. We confirm the findings of being female and the combined use of clarithromycin as risk factors for furazolidone-induced drug fever. At present, individual genetic variation in key genes involved in the metabolism, transport, or drug target can contribute to the risk of furazolidone-associated drug fever is still unclear. We want to do such studies in the future. At present, our findings can improve furazolidone-treated patients' safety to some extent. We also have added limitation in the Discussion section to clarify this in revised manuscript as follow: “Line#246-251 Secondly, drug fever may have been induced by multifactorial causes including diet, comorbidities, age, weight, drug–drug interactions, and genetics. Individual genetic variation in key genes involved in the metabolism, transport, or drug target can contribute to the risk of adverse events. Although we have examined various factors related to furazolidone-associated drug fever, some other risk factors remain unexamined which may affect this adverse event.” 

The manuscript should be checked by a native speaker in terms of language and grammar. I highlighted only some passages in the attached document.

Response: We apologize for the poor language of our manuscript. All the highlighted passages in the attached document have been revised. The manuscript has been thoroughly revised and edited by a native speaker. The certificate provided by multidisciplinary digital publishing institute was in the attached document. We really hope that the language level have been substantially improved.

I am not sure for PLOS ONE criteria for manuscript selection but the finding of the paper are not significant due to lack of data about very important variables - that can address this topic clearly

Response: Thank you for your suggestion. We agree that drug response can be impacted by several factors. In fact, our research is the first to show the independent risk factors for furazolidone-associated drug fever. We confirm the findings of being female and the combined use of clarithromycin as risk factors for furazolidone-induced drug fever. At present, our findings can improve furazolidone-treated patients' safety to some extent. As the underlying mechanisms are uncertain to date, we will do further research on this topic in the future.

Best wishes,

Yours sincerely ,

Haibin Dai

---

## [Decision Letter · Decision Letter 1]

12 Jan 2022

PONE-D-21-28795R1Risk factors of furazolidone-associated feverPLOS ONE

Dear,

Thank you for submitting your manuscript to PLOS ONE. After careful consideration, we feel that it has merit but does not fully meet PLOS ONE’s publication criteria as it currently stands. Therefore, we invite you to submit a revised version of the manuscript that addresses the points raised during the review process. Please submit your revised manuscript by 12th January 2022. If you will need more time than this to complete your revisions, please reply to this message or contact the journal office at plosone@plos.org. Please include the following items when submitting your revised manuscript:A rebuttal letter that responds to each point raised by the academic editor and reviewer(s). You should upload this letter as a separate file labeled 'Response to Reviewers'.A marked-up copy of your manuscript that highlights changes made to the original version. You should upload this as a separate file labeled 'Revised Manuscript with Track Changes'.An unmarked version of your revised paper without tracked changes. You should upload this as a separate file labeled 'Manuscript'.

We look forward to receiving your revised manuscript.

Kind regards,

Muhammad Shahzad Aslam, Ph.D.,M.Phil., Pharm-D

Academic Editor

PLOS ONE

Reviewers' comments:

Reviewer's Responses to Questions

**Comments to the Author**

1. If the authors have adequately addressed your comments raised in a previous round of review and you feel that this manuscript is now acceptable for publication, you may indicate that here to bypass the “Comments to the Author” section, enter your conflict of interest statement in the “Confidential to Editor” section, and submit your "Accept" recommendation.

2. Is the manuscript technically sound, and do the data support the conclusions?

Reviewer #1: Yes

Reviewer #2: Yes

Reviewer #3: No

3. Has the statistical analysis been performed appropriately and rigorously? 

Reviewer #1: Yes

Reviewer #2: Yes

Reviewer #3: No

4. Have the authors made all data underlying the findings in their manuscript fully available?

Reviewer #1: Yes

Reviewer #2: (No Response)

Reviewer #3: No

5. Is the manuscript presented in an intelligible fashion and written in standard English?

Reviewer #1: Yes

Reviewer #2: Yes

Reviewer #3: No

7. PLOS authors have the option to publish the peer review history of their article (what does this mean?). If published, this will include your full peer review and any attached files.

Reviewer #1: No

Reviewer #2: **Yes: **Dr. Gul Ambreen

Reviewer #3: No

6. Review Comments to the Author

Reviewer #3: 1) Data collection procedures are not clear. Which data are from medical records of patients and which are gathered from telephone calls? Did the researchers make phone calls to the 1499 patients? How was Naranja algorithm used in the study? The results of the Naranja algorithm were not shown, the researchers only kept saying about fever getting lost when the drug is discontinued. How did the researchers know that fever was lost after 48 hrs? The incidence of drug fever caused by furazolidone was 1.80% according to the article but data to prove that these are furazolidone-induced fever cases are lacking. The increase in neutrophils in the lab results of the patients was not also explained, as this could be a reason also for the fever.

2) Mean onset of fever is 11 days, how about treatment duration less than 5 days or over 14 days, where they included in the 1499 patients enrolled?

3) The number of patients with furazolidone-induced fever in the study is 27, is this statistically enough to determine the risk factors? Did you compute the number of subjects with furazolidone-induced fever needed to determine risk factors for it? Statistical analysis to show that the risk factors for furazolidone-associated fever that were female and clarithromycin were not sufficient in the study. Such as, your basis of selecting females as a risk factor compared to males is that there are 21 females and 6 males out of the 27 had the furazolidone-induced fever. There are actually more females in the 1499 enrolled patients in the study. The basis for concluding clarithromycin as a risk factor was not well discussed and was not statistically validated.

The data collection procedure was not clear. The statistical analysis and data provided to support the conclusion of the study are not sufficient.

---

## [Author Response · Author response to Decision Letter 1]

11 Feb 2022

Thank you for your letter and for the reviewers' comments concerning our manuscript entitled “Risk Factors of Furazolidone-Associated Fever”. These comments are all valuable and very helpful for revising and improving our paper, as well as the important guiding significance to our research. We have studied comments carefully and have made corrections which we hope to meet with approval. The responds to the editor and reviewers' comments are as follows: 

Reviewers' comments:

Reviewer's Responses to Questions

Comments to the Author

1. If the authors have adequately addressed your comments raised in a previous round of review and you feel that this manuscript is now acceptable for publication, you may indicate that here to bypass the “Comments to the Author” section, enter your conflict of interest statement in the “Confidential to Editor” section, and submit your "Accept" recommendation.

2. Is the manuscript technically sound, and do the data support the conclusions?

Reviewer #1: Yes

Reviewer #2: Yes

Reviewer #3: No

Response: Yes, our manuscript described a technically sound piece of scientific research. The research has been conducted rigorously, with appropriate controls and sample sizes. This retrospective cohort study identified two risk factors for furazolidone-associated fever, which were female and clarithromycin. We also analyzed the characteristics of drug fever during anti-Helicobacter pylori therapy. These conclusions were drawn appropriately based on the data presented.

3. Has the statistical analysis been performed appropriately and rigorously?

Reviewer #1: Yes

Reviewer #2: Yes

Reviewer #3: No

Response: Yes, the statistical analysis has been performed appropriately and rigorously with SPSS for Windows version 22.0 in our manuscript.

4. Have the authors made all data underlying the findings in their manuscript fully available?

Reviewer #1: Yes

Reviewer #2: (No Response)

Reviewer #3: No

Response: Yes, the data that support the findings of this study contain potentially sensitive patient information. Data are available from corresponding author Haibin Dai with the permission of the Ethics Committee of the 2nd Affiliated Hospital, School of Medicine, Zhejiang University. 

5. Is the manuscript presented in an intelligible fashion and written in standard English?

Reviewer #1: Yes

Reviewer #2: Yes

Reviewer #3: No

Response: Yes, the manuscript has been thoroughly revised and edited by a native speaker. The certificate provided by multidisciplinary digital publishing institute was in the attached document. We really hope that the language level has been substantially improved.

6. PLOS authors have the option to publish the peer review history of their article (what does this mean?). If published, this will include your full peer review and any attached files.

Do you want your identity to be public for this peer review? For information about this choice, including consent withdrawal, please see our Privacy Policy.

Reviewer #1: No

Reviewer #2: Yes: Dr. Gul Ambreen

Reviewer #3: No

7. Review Comments to the Author

Reviewer #3: 1) Data collection procedures are not clear. Which data are from medical records of patients and which are gathered from telephone calls? Did the researchers make phone calls to the 1499 patients? How was Naranja algorithm used in the study? The results of the Naranja algorithm were not shown, the researchers only kept saying about fever getting lost when the drug is discontinued. How did the researchers know that fever was lost after 48 hrs? The incidence of drug fever caused by furazolidone was 1.80% according to the article but data to prove that these are furazolidone-induced fever cases are lacking. The increase in neutrophils in the lab results of the patients was not also explained, as this could be a reason also for the fever.

Response: We feel sorry for the inconvenience brought to the reviewer. Most data in this study, for example demographic and clinical characteristics of patients, drug dosing, concomitant drugs and laboratory tests were all from medical records. The time when fever lost was gathered from telephone calls. 27 Patients with drug fever who went back to hospital for further consultation were contacted by telephone at 0.5 and 1 months after the first time of they left hospital by pharmacists. However, other 1472 patients were only contacted by telephone at about 1 month after they left hospital. 

 We feel sorry for the results of the Naranja algorithm were not shown at first. We have supplemented its method in revised manuscript as follow: “Line#109-113 Two independent reviewers gave each suspected patients with drug fever a Naranjo scale score. The scores were averaged and rounded up to the higher integer. The final score interpretations were stratified into four categories with a score of ≥ 9 considered “definite”, 5 to 8 “probable”, 1 to 4 “possible”, and those ≤ 0 “doubtful” likelihood of the drug causing the ADR”. The results of the Naranja algorithm we have supplemented in revised manuscript as follow: “Line#146-147 The Naranjo score classified 0 (0.0%) of cases as definite, 25 (92.6%) probable, 2 (7.4%) possible, and 0 (0.0%) doubtful (Table 1)”. We also have supplemented the “Table 1” in line#158 as follow: 

Table 1. Naranjo score of patients with observed fever.

Naranjo score N (%)

Definite (≥ 9) 0 (0.0)

Probable (5-8) 25 (92.6)

Possible (1-4) 2 (7.4)

Doubtful (≤ 0) 0 (0.0)

 A total of 1499 patients received furazolidone and met the overall inclusion criterion. Of these 1499 patients, 27 (1.80%) developed drug fever. We have supplemented the results of the Naranja algorithm in the revised manuscript. The Naranjo scale was used to assess the relationship between the adverse drug reactions and furazolidone therapy. Most of the score suggested a probable relationship. 

The increase in neutrophils in the lab results of the patients could be influenced by many factors, for example infection, allergy, cancer and so on. When two independent reviewers gave each suspected patients with drug fever a Naranjo scale score have considered it. Thank you for your suggestion.

2) Mean onset of fever is 11 days, how about treatment duration less than 5 days or over 14 days, where they included in the 1499 patients enrolled?

Response: Thank you for your suggestion. Our study was a retrospective case-control study of 1499 patients who used furazolidone-containing quadruple regimens for H. pylori infection. All of them took therapeutic regimen for 14 days. Of these 1499 patients, 27 patients developed drug fever. We are very sorry for haven’t stated clearly in the original version. We have revised line#93-96 to “This was a retrospective case-control study using the medical records of patients treated with 14-day Furazolidone-containing quadruple regimens for Helicobacter pylori (H. pylori) infection at the Second Affiliated Hospital of Zhejiang University, School of Medicine (SAHZU)”.

3) The number of patients with furazolidone-induced fever in the study is 27, is this statistically enough to determine the risk factors? Did you compute the number of subjects with furazolidone-induced fever needed to determine risk factors for it? Statistical analysis to show that the risk factors for furazolidone-associated fever that were female and clarithromycin were not sufficient in the study. Such as, your basis of selecting females as a risk factor compared to males is that there are 21 females and 6 males out of the 27 had the furazolidone-induced fever. There are actually more females in the 1499 enrolled patients in the study. The basis for concluding clarithromycin as a risk factor was not well discussed and was not statistically validated.

Response: Thank you for your suggestion. In fact, this was a retrospective non-interventional case-control study, the gender of enrolled patients was based on actual medical events. We also have used the tests for the odds ratio in a matched case-control design with a binary X in PASS 2021. 27 patients with furazolidone-induced fever were statistically enough to determine the risk factors. The statistical reports provided by PASS 2021 was in the attached document. 

 We feel sorry for concluding clarithromycin as a risk factor was not well discussed. As we know, the clinical characteristics of patients and concomitant medication are common risk factors of ADR. Clarithromycin is one of the concomitant drugs in enrolled patients. So, we used separate logistic regression analyses and multivariable logistic regression analyses to determine whether clarithromycin associated with drug fever in this study. Covariates were based on significant variables in the univariable model (i.e., P < 0.05). The result of separate logistic regression analyses was showed in table 2 that concomitant medications, except for amoxicillin and clarithromycin (P=0.004 and P<0.001, respectively), were similar between the patients with or without furazolidone-associated fever. The result of multivariable logistic regression analyses was showed in table 5 that two independent variables to be associated with the onset of drug fever after adjusting the underlying confounders through multivariate analysis. These variables included gender (OR, 3.162; 95% CI, 1.264-7.914; P=0.014) and clarithromycin (OR, 4.834; 95% CI, 2.165–10.794; P<0.001).

The data collection procedure was not clear. The statistical analysis and data provided to support the conclusion of the study are not sufficient.

Response: We feel sorry for the inconvenience brought to the reviewer. We have declared the data collection procedure and statistical analysis in questions 1 and 3 at above. We also have revised our manuscript try my best. Thank you for your suggestion.

Best wishes,

Yours sincerely,

Haibin Dai

---

## [Decision Letter · Decision Letter 2]

7 Mar 2022

PONE-D-21-28795R2Risk factors of furazolidone-associated feverPLOS ONE

Dear,

Thank you for submitting your manuscript to PLOS ONE. After careful consideration, we feel that it has merit but does not fully meet PLOS ONE’s publication criteria as it currently stands. Therefore, we invite you to submit a revised version of the manuscript that addresses the points raised during the review process.Please submit your revised manuscript by 5 April 2022. If you will need more time than this to complete your revisions, please reply to this message or contact the journal office at plosone@plos.org. Please include the following items when submitting your revised manuscript:A rebuttal letter that responds to each point raised by the academic editor and reviewer(s). You should upload this letter as a separate file labeled 'Response to Reviewers'.A marked-up copy of your manuscript that highlights changes made to the original version. You should upload this as a separate file labeled 'Revised Manuscript with Track Changes'.An unmarked version of your revised paper without tracked changes. You should upload this as a separate file labeled 'Manuscript'.If applicable, we recommend that you deposit your laboratory protocols in protocols.io to enhance the reproducibility of your results. Protocols.io assigns your protocol its own identifier (DOI) so that it can be cited independently in the future. For instructions see: https://journals.plos.org/plosone/s/submission-guidelines#loc-laboratory-protocols. Additionally, PLOS ONE offers an option for publishing peer-reviewed Lab Protocol articles, which describe protocols hosted on protocols.io. Read more information on sharing protocols at https://plos.org/protocols?utm_medium=editorial-email&utm_source=authorletters&utm_campaign=protocols.

We look forward to receiving your revised manuscript.

Kind regards,

Muhammad Shahzad Aslam, Ph.D.,M.Phil., Pharm-D

Academic Editor

PLOS ONE

Journal Requirements:

Reviewers' comments:

Reviewer's Responses to Questions

**Comments to the Author**

1. If the authors have adequately addressed your comments raised in a previous round of review and you feel that this manuscript is now acceptable for publication, you may indicate that here to bypass the “Comments to the Author” section, enter your conflict of interest statement in the “Confidential to Editor” section, and submit your "Accept" recommendation.

Reviewer #1: All comments have been addressed

Reviewer #2: (No Response)

Reviewer #4: All comments have been addressed

2. Is the manuscript technically sound, and do the data support the conclusions?

Reviewer #1: Yes

Reviewer #2: Yes

Reviewer #4: Partly

3. Has the statistical analysis been performed appropriately and rigorously? 

Reviewer #1: Yes

Reviewer #2: Yes

Reviewer #4: Yes

4. Have the authors made all data underlying the findings in their manuscript fully available?

Reviewer #1: Yes

Reviewer #2: (No Response)

Reviewer #4: Yes

5. Is the manuscript presented in an intelligible fashion and written in standard English?

Reviewer #1: Yes

Reviewer #2: Yes

Reviewer #4: Yes

6. Review Comments to the Author

Reviewer #1: All the comments have been addressed

The data collection procedure is clear. The statistical analysis and data provided

to support the conclusion of the study are sufficient.

Reviewer #2: (No Response)

Reviewer #4: . Congratulation to the author and team. It is an intersting findings particluarly in pharmacovigillence. Please refer to the attached comments.

7. PLOS authors have the option to publish the peer review history of their article (what does this mean?). If published, this will include your full peer review and any attached files.

Reviewer #1: **Yes: **Sadia Shakeel

Reviewer #2: **Yes: **Dr. Gul Ambreen

Reviewer #4: No

---

## [Author Response · Author response to Decision Letter 2]

24 Mar 2022

Thank you for your letter and for the reviewers' comments concerning our manuscript entitled “Risk Factors of Furazolidone-Associated Fever”. These comments are all valuable and very helpful for revising and improving our paper, as well as the important guiding significance to our research. We have studied comments carefully and have made corrections which we hope to meet with approval. The responds to the editor and reviewers' comments are as follows:

Journal Requirements: 

Response: Thank you for your suggestion. We have reviewed the format of references according to the authors' instructions of PLOS ONE and modified one by one according to it. We use the relevant current reference replace of the original reference 2, because the original reference is a Chinese literature, which cannot be found on the web of science or PubMed. The other reference we all have added the PMID numbers and made appropriate modifications to ensure that references are complete and correct.

Reviewers' comments:

Reviewer's Responses to Questions

Comments to the Author

1. If the authors have adequately addressed your comments raised in a previous round of review and you feel that this manuscript is now acceptable for publication, you may indicate that here to bypass the “Comments to the Author” section, enter your conflict of interest statement in the “Confidential to Editor” section, and submit your "Accept" recommendation.

Reviewer #1: All comments have been addressed

Reviewer #2: (No Response)

Reviewer #4: All comments have been addressed

2. Is the manuscript technically sound, and do the data support the conclusions?

Reviewer #1: Yes

Reviewer #2: Yes

Reviewer #4: Partly

Response: Yes, our manuscript described a technically sound piece of scientific research. The research has been conducted rigorously, with appropriate controls and sample sizes. This retrospective case-control study identified two risk factors for furazolidone-associated fever, which were female and clarithromycin. We also analyzed the characteristics of drug fever during anti-Helicobacter pylori therapy. These conclusions were drawn appropriately based on the data presented.

3. Has the statistical analysis been performed appropriately and rigorously?

Reviewer #1: Yes

Reviewer #2: Yes

Reviewer #4: Yes

4. Have the authors made all data underlying the findings in their manuscript fully available?

Reviewer #1: Yes

Reviewer #2: (No Response)

Reviewer #4: Yes

Response: Yes, the data that support the findings of this study contain potentially sensitive patient information. Data are available from corresponding author Haibin Dai with the permission of the Ethics Committee of the 2nd Affiliated Hospital, School of Medicine, Zhejiang University.

5. Is the manuscript presented in an intelligible fashion and written in standard English?

Reviewer #1: Yes

Reviewer #2: Yes

Reviewer #4: Yes

6. Review Comments to the Author

Reviewer #1: All the comments have been addressed

The data collection procedure is clear. The statistical analysis and data provided

to support the conclusion of the study are sufficient. 

Reviewer #2: (No Response) 

Reviewer #4: Congratulation to the author and team. It is an intersting findings particluarly in pharmacovigillence. Please refer to the attached comments. 

Reviewer #4: 1) It is mentioned that it was a retrospective case-control study, and it is also mentioned that the data was from July to September 2018. – When the study was conducted? When did the investigators called the subjects. 2019? / 2020?/ 2021? – This is to evaluate the reliability of the collected data on potential recall biases. Thank you.

Response: Thank you for your suggestion. We feel sorry for the inconvenience brought to the you. The fact was that from July to September 2018, doctors in the fever clinic of our hospital reported many adverse reactions of drug fever caused by furazolidone (27 cases in total). When there are more than three cases of the same adverse reaction of the same drug per month, an investigation should be initiated in our hospital. I was one of the supervisors of adverse drug reactions in our hospital. The work of ADR supervisor was that we need to find out the causes of drug fever aggregation caused by furazolidone, track the outcomes of ADR and ask other patients used furazolidone whether they have febrile reactions. We changed use of furazolidone with different production batch numbers, checked the circulation and storage conditions of furazolidone, but found no possible causes of drug fever. After excluding drug quality, drug circulation, drug storage and other reasons, we considered looking for other risk factors of furazolidone-associated fever. We called the patients in 2018 and the study was also conducted in 2018. However, we are usually very busy with our work and not very proficient in English, this article has not be completed until 2021.

2) Line 104 – did you exclude immunocompromised patients?

Response: Thank you for your suggestion. We feel sorry that we only excluded the following patients (1) patients who were pregnant or breastfeeding; (2) patients who received furazolidone treatment not for Helicobacter pylori infection; (3) patients taking antibiotics or any acid suppressant or non-steroidal anti-inflammatory drug in the last 4 weeks; and (4) patients with chronic hepatic, renal, or pulmonary disease. Though immunocompromised patients did not meet the exclusion criteria of this study, we may have excluded partial of immunocompromised patients according to the present criteria. 

3) Line 113- Fever was defined as body temperature ≥ 38°C – How did the investigator know that the body temperature of the subjects (who reported as having drug fever) ≥ 38°C? Can you explain in methodology how the investigator gets the information when the nature of the study was retrospective and patients were treated as out-patient. The investigators called the subjects? How long a part? How you minimize recall biases? Who scanned the body temperature? 

Response: Thank you for your suggestion. I am one of the adverse drug reaction supervisors in our hospital. First of all, the doctors in the fever clinic reported adverse drug reaction of furazolidone-associated fever to us. Our hospital uses an in-hospital ADR reporting system, its interface as shown in the below. We got the general situation of adverse drug reactions from the hospital reporting system, and then got more information from medical records, such as body temperature. The 27 patients' body temperature was scanned by nurses. ADR supervisors telephoned the patients. One patient communicated by telephone for about 3 minutes. Three of the investigators in this study were ADR supervisors who were responsible for furazolidone-associated fever in 2018. ADR supervisors telephoned patients using furazolidone was documented on paper, which could avoid the recall biases of ADR supervisors. However, most patients received telephone follow-up one month after dispensing, and these patients may have some recall biases. The effects of recall bias we have mentioned in the limitations of this study, see lines 254-256. Thank you. 

The picture of the interface for hospital ADR reporting system can be seen in the attachment of the response to reviewers.

4) Line 115 – “ we conducted a follow up assessment…” – But this was a retrospective study… unless it was a retrospective, prospective 2 phases study. Appreciate if the author can clarify.

Response: Thank you for your suggestion. We feel sorry for the inconvenience brought to the reviewer. We have shown in questions 1 and 3 that this study evolved during the investigation of the adverse reaction of furazolidone-associated fever. We did not attempt to analyze other risk factors until the end of the adverse reaction investigation when no other causes could be found. So, we think this was a retrospective study. Thank you. 

5) Line 122 – how the investigators obtained the WBC, Neutrophils and eosinophils? Did the subjects admitted to ward? Unless this was a prospective interventional study that specified the protocol to obtain blood investigations at specified date. Can the author please clarify?

Response: Thank you for your suggestion. 27 Patients with fever who went back to hospital were assigned to the fever clinic for further consultation. Patients in the fever clinic will have routine blood tests. Routine blood tests were performed in all the 27 patients in the fever clinic. We collected their laboratory tests results from medical records. 27 patients with fever were only treated in the fever clinic and they did not admit to ward. Blood routine test was a routine examination in the fever clinic, not a clinical intervention in our study.

6) The most important is the author able to convince the readers how reliable is the data? How comparable the control to case. Once this part is clarified the result and discussion they are fine to me. 

Response: We feel sorry for the inconvenience brought to the you. We have described the process of this study in detail in questions 1 to 5 at above. We really hope that we have clarified clearly to convince the readers. Thank you for your suggestion.

7. PLOS authors have the option to publish the peer review history of their article (what does this mean?). If published, this will include your full peer review and any attached files.

Do you want your identity to be public for this peer review? For information about this choice, including consent withdrawal, please see our Privacy Policy.

Reviewer #1: Yes: Sadia Shakeel

Reviewer #2: Yes: Dr. Gul Ambreen

Reviewer #4: No

Best wishes,

Yours sincerely,

Haibin Dai

---

## [Decision Letter · Decision Letter 3]

28 Mar 2022

Risk factors of furazolidone-associated fever

PONE-D-21-28795R3

Dear,

We’re pleased to inform you that your manuscript has been judged scientifically suitable for publication and will be formally accepted for publication once it meets all outstanding technical requirements.

Kind regards,

Muhammad Shahzad Aslam, Ph.D.,M.Phil., Pharm-D

Academic Editor

PLOS ONE

Additional Editor Comments (optional):

Reviewers' comments:

Reviewer's Responses to Questions

**Comments to the Author**

1. If the authors have adequately addressed your comments raised in a previous round of review and you feel that this manuscript is now acceptable for publication, you may indicate that here to bypass the “Comments to the Author” section, enter your conflict of interest statement in the “Confidential to Editor” section, and submit your "Accept" recommendation.

Reviewer #4: All comments have been addressed

2. Is the manuscript technically sound, and do the data support the conclusions?

Reviewer #4: Yes

3. Has the statistical analysis been performed appropriately and rigorously? 

Reviewer #4: Yes

4. Have the authors made all data underlying the findings in their manuscript fully available?

Reviewer #4: Yes

5. Is the manuscript presented in an intelligible fashion and written in standard English?

Reviewer #4: Yes

6. Review Comments to the Author

Reviewer #4: (No Response)

7. PLOS authors have the option to publish the peer review history of their article (what does this mean?). If published, this will include your full peer review and any attached files.

Reviewer #4: No

---

## [Editor Report · Acceptance letter]

30 Mar 2022

PONE-D-21-28795R3 

Risk factors of furazolidone-associated fever 

Dear Dr. Dai:

I'm pleased to inform you that your manuscript has been deemed suitable for publication in PLOS ONE. Congratulations! Your manuscript is now with our production department. 

Kind regards, 

on behalf of

Dr. Muhammad Shahzad Aslam 

Academic Editor

PLOS ONE